# Controlling the Surface Roughness of Surface-Electrode Ion Trap Based on Micro-Nano Fabrication

Yizhu Hou [1,†] , Xinfang Zhang [2,3,†] , Wei Wu [1], Ting Zhang [1], Pingxing Chen [1] and Zhijiao Deng [1,*]

1  Department of Physics, College of Liberal Arts and Sciences, National University of Defense Technology, Changsha 410073, China; houyizhu@nudt.edu.cn (Y.H.); weiwu@nudt.edu.cn (W.W.); tingzhang@nudt.edu.cn (T.Z.); pxchen@nudt.edu.cn (P.C.)
2  College of Computer Science and Technology, National University of Defense Technology, Changsha 410073, China; xfz-6446@nudt.edu.cn
3  State Key Laboratory of High Performance Computing, National University of Defense Technology, Changsha 410073, China
*  Correspondence: dengzhijiao@nudt.edu.cn
†  These authors contributed equally to this work.

**Abstract:** The surface-electrode ion trap is one of the most promising devices to realize large-scale and integrated quantum information processing. However, a series of problems are faced in the micro-nano fabrication of surface-electrode ion traps. A prominent one is the difficulty to control the thick film surface roughness. A rough electrode surface could produce excessive radio frequency (RF) loss and deteriorate trapping ability of the surface-electrode ion trap. In this paper, a thick film micro-nano fabrication technology to control the surface roughness is presented, which can reduce the roughness of thick film surface-electrode down to 6.2 nm, while being controllable between 6.2 nm and 45 nm. Therefore, it can also provide a basis for studying the influence of electrode surface roughness on trap performance. The micro-nano fabrication technology is not only suitable for surface-electrode ion traps with various configurations, but also be further applied to researches of MEMS, solar cells and surface science.

**Keywords:** surface-electrode ion trap; micro-nano fabrication; thick film; surface roughness

## 1. Introduction

The trapped ion system is one of the candidates for realizing a quantum computer [1,2]. The microfabricated traps are used to scale up [3], integrate [4] and miniaturize trapped ion system [5]. Both surface-electrode ion trap (SEIT) [6] and 3D microscopic ion trap [7] are microfabricated traps, while the structure and fabrication of SEIT are simpler. At present, SEIT already has been widely applied to quantum simulation [8] and quantum information processing [9].

A well designed and fabricated SEIT should be able to trap ions steadily with a low heating rate, and shuttle ions without exciting their motional states [10]. Currently, the methods for designing SEIT have been well established [11]. However, the micro-nano fabrication technologies of SEIT are not mature yet, mainly manifested in the difficulty to control the surface roughness of SEIT during the fabrication process [12]. The electrode fabrication of SEIT mainly includes two categories, i.e., the thin film [13] and the thick film [14]. The thick film fabrication is focused in the paper for two reasons. First, the electrode of SEIT must be thick enough to withstand high RF voltage in the trapped ion experiments [15]. Secondly, thick-film electrode allows a larger distance between the ions and the substrate, so the ion heating rate could be effectively suppressed, as the influence of the charged substrate on trapping ions is reduced [16].

The ions are trapped by the pseudopotential generated by the electrodes of SEIT. Thus, the electrode surface roughness will definitely affect the performance of SEIT, such as ion

trapping, ion manipulation, quantum gate fidelity, and ion heating rate [17]. A simpler explanation is that reducing surface roughness would bring a more uniform electric field and less surface adsorbate, which will reduce the electrical noise due to the electrode surface [18]. However, both quantitative relation and intrinsic mechanism of such effect are not clear yet. Not only the theoretical research is almost blank, but also the experimental study is rare due to the challenge of fabricating thick film SEITs with controllable surface roughness [19]. Therefore, to fabricate an electrode with controllable surface roughness is vital to the experiments with SEIT.

The control of surface roughness is actually a common problem faced in many research fields. The arithmetical mean deviation $Ra$ is a general metric of roughness. It is defined as $Ra = \frac{1}{n} \sum_{i=1}^{n} |y_i - A|$, where $n$ represents the number of measurements, $y_i$ represents the $i$-th measured height, and $A$ represents the average height of surface sampling points [20]. $Ra$ is around 50 nm for most thick-film SEIT electrode surface [15,21]. Besides, the surface roughness of electroplated Au films for connector contact materials is $Ra = 62$ nm [22], while $Ra = 10.7$ nm for ultra-fine grained Au films with high compressive strength [23]. There is also special technology in surface scientific field which could achieve several nanometers roughness [24,25]. However, the specialized equipment required is inaccessible for most SEIT research [26]. Usually in-situ argon-ion-beam cleaning [27], pulsed-laser cleaning [28] and other methods are used to clean the electrode surface of SEIT to reduce negative influence which is caused by the surface roughness. However, these methods are far from enough to achieve ideal surface roughness. In the paper, the research on reducing electrode surface roughness at its source is carried out and a new fabrication process of SEIT is proposed, which can control and reduce the surface roughness. This micro-nano fabrication process can also be applied to thick film SEITs with other geometry [29,30].

The paper is organized as follows. Section 2 gives the working principle of SEIT, analyzes and verifies a new fabrication process. Then in Section 3, the electroplating was systematically designed and optimized. The relationship between pulse frequency, electroplating solution temperature and electrode surface roughness under pulse electro-plating was discussed. The fabrication of SEITs with various roughness can be achieved by carefully adjusted electroplating process. The SEIT fabricated in this work achieved a low roughness $Ra = 6.2$ nm, and $Ra$ can be controlled to change within 6.2 nm to 45 nm by varying the electroplating parameters. Conclusion is presented in Section 4.

## 2. Method and Fabrication

The research on the micro-nano fabrication process of thick film SEIT is based on a symmetrical five-wire SEIT (Figure 1a). The SEIT includes two 145 μm-wide radio frequency electrodes, one 180 μm-wide ground electrode and multiple 150 μm-wide DC electrodes. The pseudopotential distribution is generated by two radio frequency electrodes (Figure 1b). According to this distribution, it can be known that the ions are trapped at 100 μm above the surface of SEIT (Figure 1c). In order to obtain the ideal electric potential field like Figure 1b, it is very important to reduce the surface roughness of SEIT by micro-nano fabrication.

### 2.1. Thick Film Micro-Nano Fabrication Process with Controllable Surface Roughness

Seidelin et al. [14] and Ou et al. [15] have already used the conventional thick film fabrication process in ion traps research, which includes magnetron sputtering coating, patterning lithography, electroplating and wet etching. In the etching process, the etching solution can not only etch the seed Au layer, but also etch the Au electrode surface. These will make electrode surface roughness increase dramatically. To compare the electrode surface roughness before and after etching, at least three specimens were fabricated under each test condition, the roughness profile of all specimens has been rated, and the results are very close, one of them is shown in Figure 2. Figure 2a,d demonstrates the scanning images of scanning electron microscope (SEM Device: S-4800, HITACHI, Tokyo, Japan). Through the comparison of SEM images, it can be clearly seen that the electrode surface

becomes much rougher after etching. In order to further quantify the electrode surface roughness, atomic force microscope (AFM) was used (Device: NTEGRA, SPM Mode: Semicontact Topography, Scan velocity: 10.12 μm/s, Step: 19.61 nm, NT-MDT, Moscow, Russia). The electrode surfaces are scanned by AFM in 5 μm × 5 μm region, each profile has 65536 sampling points of height (Figure 2b,e). The statistical diagrams of the height of the sampling point are shown in Figure 2c,f. The scanned data is processed by Nova software (NT-MDT, v1.1.0.1918, Moscow, Russia) to determine $Ra$, whose values before and after etching are 9.6 nm and 54.9 nm, respectively. This shows that wet etching has a great influence on the surface roughness.

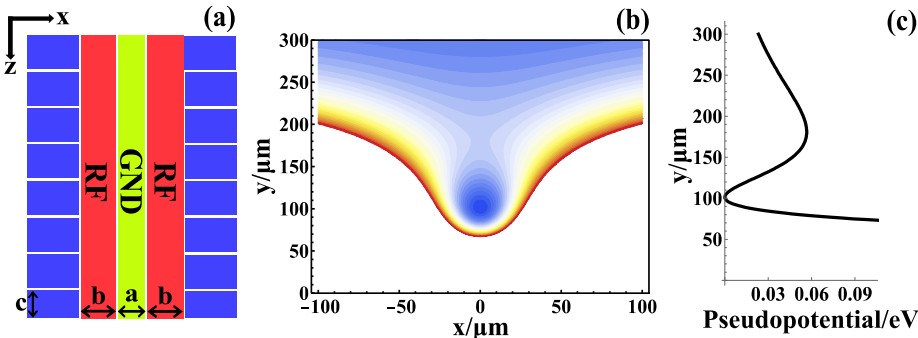

**Figure 1.** Symmetrical five-wire SEIT: (**a**) The geometric dimensions of the electrode. The widths of the ground electrode, RF electrode and segmented DC control electrode are $a$ = 180 μm, $b$ = 145 μm, and $c$ = 150 μm, respectively; (**b**) The radial distribution of pseudopotential. The pseudopotential decreases from red to blue; (**c**) Pseudopotential distribution in the y direction ($x$ = $z$ = 0). The trap depth is about 0.06 eV. The trapping height and escaping point is about 100 μm and 175 μm, respectively.

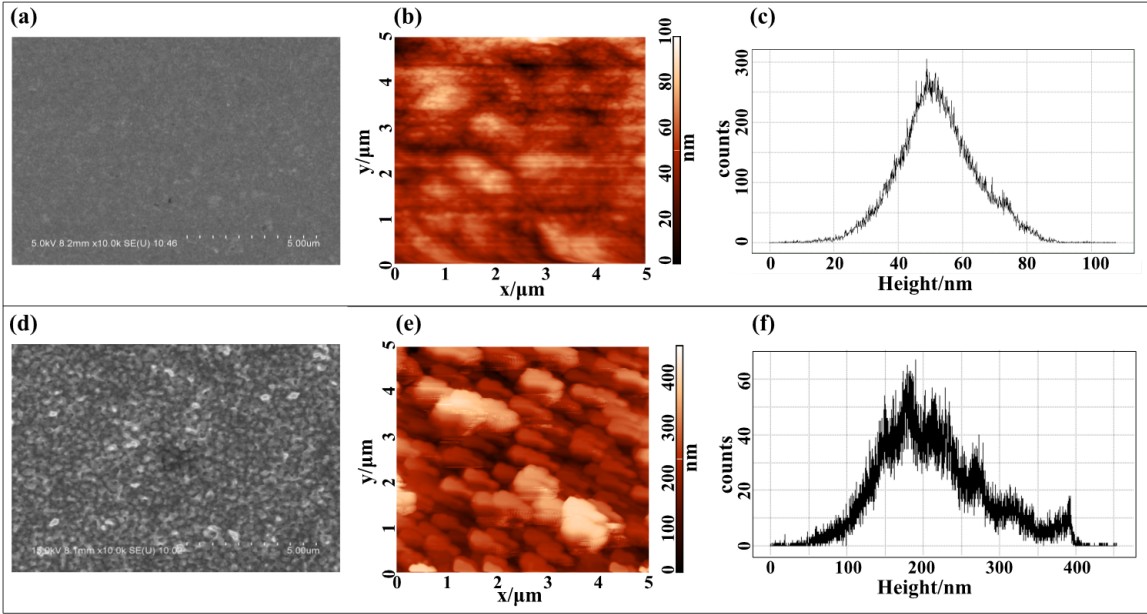

**Figure 2.** The surface of the electrode before and after etching without protective layer. SEM images before etching (**a**) and after etching (**d**); AFM images before etching (**b**) and after etching (**e**); The counts of the height of the sampling point before etching, $Ra$ = 9.6 nm (**c**) and after etching, $Ra$ = 54.9 nm (**f**).

To protect the Au thick film electrode surface from wet etching, a new proposal to optimize the thick film micro-nano fabrication process is given in Figure 3. Based on the routine thick film fabrication process, a 10 nm titanium film has been magnetron sputtered as a protective layer after electroplating. The titanium has been chosen for two reasons. On

the one hand, the titanium protective layer and the bonding layer are completely etched in the process of the Figure 3h,i. On the other hand, comparing with photoresist or nonmetallic materials, there will be no nonmetallic residue or influence from the distribution of surface charge on the electrode.

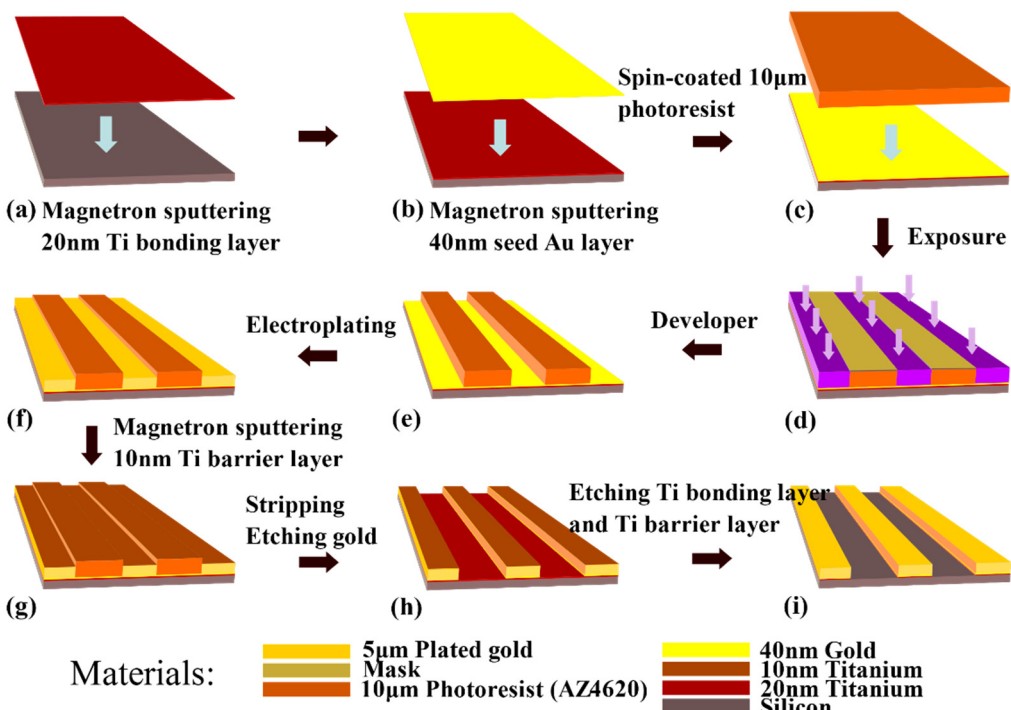

**Figure 3.** Thick film micro-nano fabrication process with controllable surface roughness.

The comparison of the electrode surface roughness before and after etching with protective layer is given in Figure 4. The SEM images and AFM images are shown in Figure 4a,d and Figure 4b,e. It can be seen that there is no obvious change of the electrode surface. The statistical diagrams of the height of the sampling point are shown in Figure 4c,f. The scanned data is processed by Nova software to determine *Ra*, whose values are 8.9 nm and 6.2 nm (Both values are achieved under the optimized electroplating conditions explained in Table 1. (Section 3)) before and after etching, respectively. It shows that the titanium protective layer is very effective in protecting the electrode surface during etching. So, the fabrication of SEITs with different roughness can be effectively controlled by the electroplating process.

**Table 1.** Optimized pulse electroplating parameters.

| Parameter Name | Value |
| --- | --- |
| Pulse peak current density $i_p$ | 5 A/dm$^2$ |
| Duty ratio $\lambda$ | 1:9 |
| Pulse frequency $f$ | 200 Hz |
| Electroplating solution temperature $T$ | 40 °C |
| Pulse electroplating time | 366 s |
| Distance between cathode and anode | 80 mm |

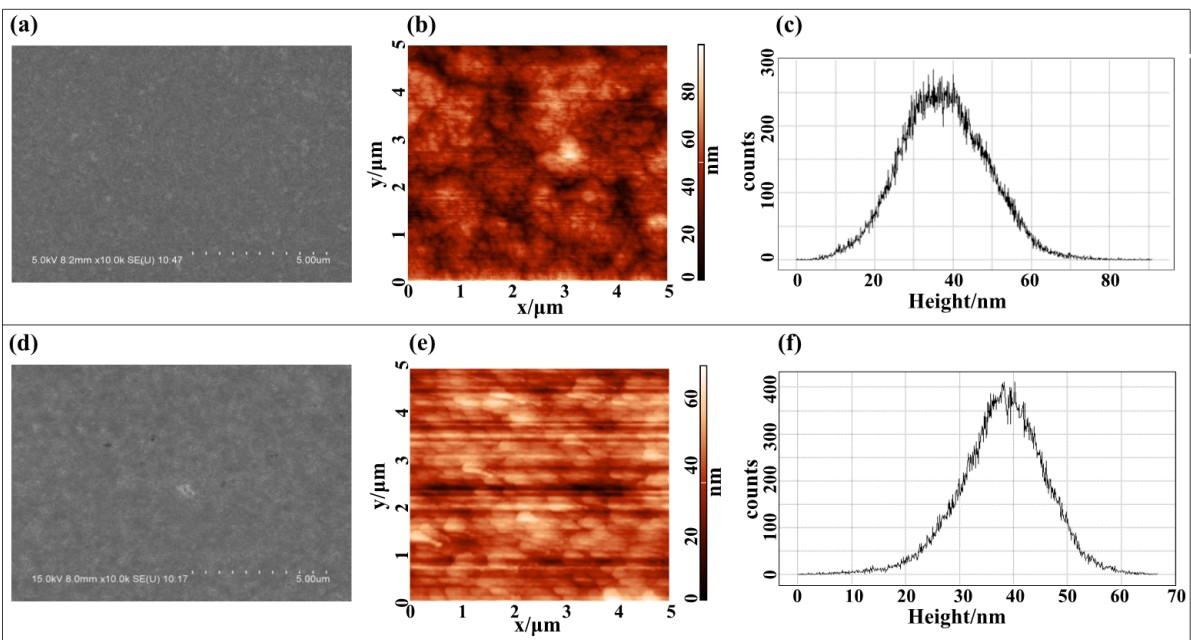

**Figure 4.** The surface of the electrode before and after etching with protective layer. SEM images before etching (**a**) and after etching (**d**); AFM images before etching (**b**) and after etching (**e**); The counts of the height of the sampling point before etching, $Ra = 8.9$ nm (**c**) and after etching, $Ra = 6.2$ nm (**f**).

### 2.2. Fabrication of Thick Film Surface-Electrode Ion Trap with Different Roughness

The experimental results of thick film micro-nano fabrication process show that electroplating is the major influence on the surface roughness. Therefore, the relationship between electroplating parameters and electrode surface roughness has been analyzed to optimize the fabrication of SEIT.

The methods of electroplating are diverse [31], mainly including DC electroplating and pulse electroplating. DC electroplating is widely used nowadays in SEIT research [15,21], but the controllability of DC electroplating is poor because there are only two parameters for DC electroplating, i.e., current density and electroplating time. By contrast, the parameters of pulse electroplating include pulse duration $t_{on}$, duty ratio $\lambda = t_{on}/t_{off}$, pulse peak current density $i_p$, and average current density $i_m = (\lambda \cdot i_p)/(1 + \lambda)$. Meanwhile, the pulse electroplating has many advantages. It yields more compact coating structure, smaller grain size, lower porosity, and fewer impurities [23]. Thus, there is more room for the optimization of pulse electroplating, which is exactly to be used in the following.

In the electroplating experiment, the cathode substrate is a P-type heavily doped silicon wafer (Nanjing MKNANO Tech. Co., Ltd., Nanjing, China), on which there are in turn 20 nm titanium bonding layer, 40 nm seed golden layer, and a pattern structure with a thickness of 10 µm. The area of cathode and anode is 3 cm$^2$ and 30 cm$^2$, respectively. The distance between the cathode and anode is 80 mm. The gold electroplating solution used is a sulfite solution (METALOR® ECF-88K).

Pulse peak current density $i_p = 5$ A/dm$^2$, duty ratio $\lambda = 1{:}9$ are the proper parameters for thick film electrode fabrication, also strong stirring is needed. The thickness of the electrode is determined by [32],

$$t = adh/ZI \tag{1}$$

where $t$ the electroplating time, $a$ the cathode electroplating area, $d$ the gold density, $h$ the target electrode thickness, $Z$ the electrochemical equivalent, and $I$ the cathode current. The film thickness of the target electrode in this experiment is 5 mm, and the electroplating time used is 366 s based on the calculation of the above formula.

## 3. Results and Discussion

In this section, the following parameters are fixed in this experiment: Pulse peak current density $i_p$ = 5 A/dm$^2$, duty ratio $\lambda$ = 1:9. Experiments show that the electrode surface roughness can be significantly reduced by optimizing the pulse frequency $f$ and electroplating solution temperature $T$, so these two parameters are mainly discussed in this section. At least three specimens were fabricated under each test condition and each specimen was characterized four times by AFM.

As shown in Figure 5, the roughness of the thick film electrode decreases significantly with the pulse frequency $f$ increasing from 166.7 Hz to 333.3 Hz (pulse duration $t_{on}$ = 0.3 ms) at 20 °C. Then, the electrode surface roughness begins to increase until the pulse frequency reaches to 1000 Hz. At 40 °C, the relationship between roughness and frequency is similar. The electrode surface roughness begins to increase when the pulse frequency reaches to 200 Hz.

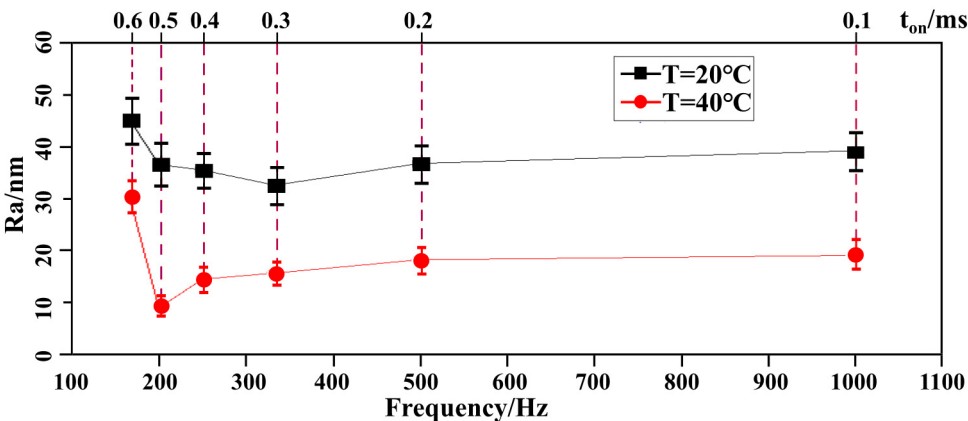

**Figure 5.** The relationship between *Ra* and the frequency of pulse electroplating at 20 °C and 40 °C.

During one pulsed electroplating cycle, gold atoms are deposited on the cathode. With the consumption of ions near the cathode, there is a concentration difference in the solution. It makes the ions move toward the cathode and leads to a competition between consumption and replenishment of ions near the cathode. Different pulse parameters yield different patterns of ion concentration near the cathode. Keep the temperature and duty ratio unchanged, then $t_{on}$ and $t_{off}$ will increase (decrease) with the pulse frequency $f$ decreasing (increasing). If there are always enough ions near the cathode throughout the pulse, a large number of gold atoms will be deposited on the surface of the cathode. As time passes by, the distribution of gold atoms on the electrode surface tends to be uniform. A low-roughness electrode would be formed. When the pulse frequency $f$ is too low, the ion concentration near the cathode can't recover quickly enough. Then it is more likely to get a high-roughness electrode. However, when the pulse frequency $f$ is too high, the number of gold atoms deposited on the cathode within one pulse period is too small, the distribution of gold atoms on the electrode surface can hardly be uniform. This also can increase the electrode surface roughness.

Moreover, the surface roughness of the thick film electrode is related to the temperature of the electroplating solution during the pulse electroplating, which is shown in Figure 6. It can be seen from Figure 6 that the order of *Ra* is $Ra_{40°C} < Ra_{60°C} < Ra_{20°C}$. With the temperature increasing above room temperature, the solubility of the salt increases so that anode passivation is prevented, then the conductivity of the solution also increases so that the dispersion ability of the electrolyte is improved. The series of changes eventually lead to the reduction of the electrode surface roughness. However, if the temperature increases beyond a critical value (about 40 °C under the specific condition shown in Figure 6, the activation energy of the discharged ions near the cathode will increase. Then the electrochemical polarization is reduced and the electrode surface roughness increases [33].

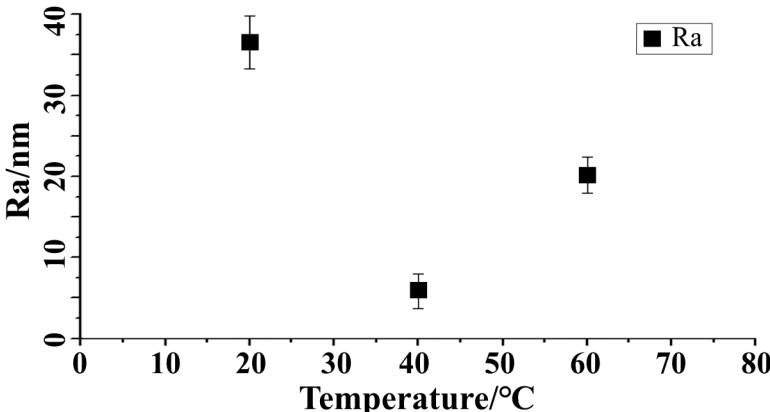

**Figure 6.** The relationship between *Ra* and electroplating solution temperatures with the pulse frequency fixed at 200 Hz.

The pulse electroplating parameters are optimized through many experiments and shown in Table 1. A bunch of thick-film SEITs were fabricated under the optimized conditions. The overall surface profile is observed and characterized by the metallographic microscope, SEM and AFM (Figure 7a,b). Because the measuring range of AFM is in nanometers, a step device was used to measure the thickness of the electrode. It is about 4.16 μm (Figure 7c).

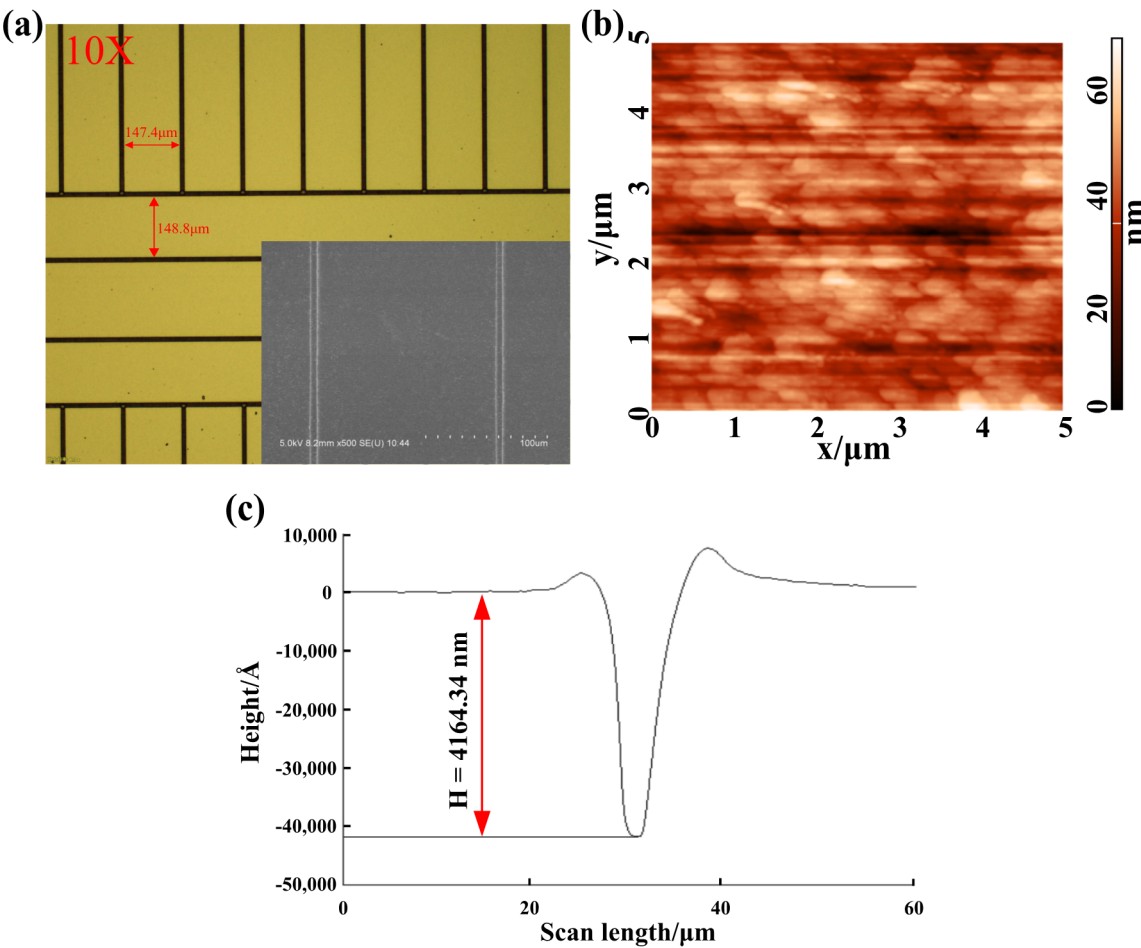

**Figure 7.** (**a**) Overall surface of thick film SEIT, the illustration is a SEM image; (**b**) The *Ra* is 6.2 nm which is characterized by AFM; (**c**) The thickness of the electrode is 4.16 μm which is measured by a-step device.

The SEIT fabrication process with a protective layer can greatly reduce the negative impact of wet etching on the surface roughness. In addition, the relationship between *Ra* and pulse frequency and electroplating solution temperature is studied. The experimental results show that there is a set of optimal pulse frequency and solution temperature under which the lowest surface roughness could be achieved. The general optimization process of pulse electroplating can be divided into two steps: First, the variation of *Ra* with temperature is experimentally calibrated in order to find an appropriate temperature; Then, the relationship of *Ra* with pulse frequency is experimentally studied under the appropriate temperature. By this calibration method, not only the minimum surface roughness can be obtained, but also the electrode surface roughness can be controlled within a certain range by changing the solution temperature and pulse frequency of the pulse electroplating. *Ra* can be reduced to 6.2 nm under the optimal temperature of 40 °C and frequency of 200 Hz (Table 1). Meanwhile, the controllable range of *Ra* is between 6.2 nm and 45 nm. At present, *Ra* of SEIT with thick film used in ion traps research is about 50 nm [15,21], and the influence of experimental parameters on the surface roughness of SEIT has never been discussed. In this work, the surface roughness is reduced by an order of magnitude, and how the electroplating parameters affect the electrode surface roughness is discussed in detail. The proposed thick film micro-nano fabrication can also be used to study the influence of electrode surface roughness on abnormal heating of ions. Most importantly, the electrode with low surface roughness can reduce the RF loss of the electrode, thereby improve the performance of SEIT for quantum computation. Moreover, the fabrication technology is simple and effective, far superior to the complicated surface cleaning techniques currently used [27,28]. Chen et al. [23] compared the effects of DC electroplating and pulse electroplating and found that pulse electroplating is better than DC electroplating, *Ra* can achieve 10.7 nm, while this paper mainly discusses optimization of pulse electroplating parameters and obtains a lower surface roughness *Ra* = 6.2 nm.

## 4. Conclusions

A thick film micro-nano fabrication process with controllable surface roughness is proposed in the paper, which overcomes the difficulty to control the surface roughness. By adjusting the pulse frequency and solution temperature during the pulse electroplating, *Ra* can be reduced to 6.2 nm, and *Ra* within 6.2 nm to 45 nm can be achieved. Through this controllable process, the relationship between the roughness and the heating rate of ions and the trapping stability of SEIT can be further studied experimentally, providing basis for chip fabrication in precise ion trap quantum computing. The thick film micro-nano fabrication process is not only suitable for surface-electrode ion traps with various configurations, but also can be further applied to researches of MEMS, solar cells and surface science.

**Author Contributions:** Conceptualization, Y.H., X.Z., Z.D., W.W., and P.C.; Experiment: Y.H. and X.Z.; Characterization analysis: Y.H. and X.Z.; Writing—original draft preparation: Y.H. and T.Z.; Writing—review and editing, all authors; Funding acquisition: W.W. and Z.D. All authors have read and agreed to the published version of the manuscript.

**Funding:** This research was funded by the National Basic Research Program of China under Grant No. 2016YFA0301903, the National Natural Science Foundation of China under Grant Nos. 11574398, 11904402, and 61632021.

**Institutional Review Board Statement:** Not applicable.

**Informed Consent Statement:** Not applicable.

**Data Availability Statement:** Data is contained within the article.

**Conflicts of Interest:** The authors declare no conflict of interest.

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
