# Peer review of "Controlling the Surface Roughness of Surface-Electrode Ion Trap Based on Micro-Nano Fabrication"

_coatings, doi:10.3390/coatings11040406_

Round 1

Reviewer 1 Report

The current paper introduce a new method to control the surface roughness using a thick microfilm nano fabrication. The aim here is to reduce the surface roughness and study the influence of the electrode surface roughness on the trap performance.

First of all the paper is missing numbers in each page so it is difficult to point out my comments location but I will try my best to explain that.

In the introduction please avoid using bulk citations unless they are given full credit in the following sections. For example references 4-8 and 11-14.

The authors cited 17 papers in a paragraph of 6 lines!

Please avoid using we in the manuscript, check everywhere and change.

Figure 1 does not belong to the introduction and should be moved to the methods and fabrication section

Figure 2 can be removed as this is a common knowledge in the micro nano fabrication of surface electrodes (suggested)

How many specimens were fabricated under each test condition?

Remove figure 6 it does not add any value to the manuscript or combine it with previous figures.

Figure 8 add error bars to the roughness data

Why it makes deposition of gold more difficult ?

There is no discussion at all in this manuscript and comparison with past work similar to this one or close to it. The authors must provide critical discussion of their findings and explain them and correlate them to past studies when possible.

The authors should consider adding some SEM images to show the surface before and after fabrication. It is least expected to have them to give better idea to the surface condition

Author Response

Dear Reviewer,

Thank you very much for your valuable comments and revision opinions. We have made great changes to the manuscript according to the comments. Please see the attachment. 

Reviewer 2 Report

Dear Authors,

In paragraph 2.1, at the end you state "The surface roughness and the average roughness (Ra) of 25 um2 square zone are depicted in Figure 3". How was the parameter Ra measured which measures the surface roughness profile? How many profiles you rated? What were the measurement parameters and methodology? Please complete this information. Figures 3 and 4 show "The average surface roughness is about…" By this you mean the parameter Ra? Because the parameter Ra expresses the mean arithmetic deviation of the roughness profile, not the average value. The conclusion is quite vague, please describe the results achieved in more detail.

Author Response

(The authors gave the same response as above.)

Reviewer 3 Report

In page 2/9, line 7, a space is missing between value and unit. Please correct.

Usually scientific texts are written in the third person.  The authors schoosen “we” and May be this is acceptable

Author Response

(The authors gave the same response as above.)

Round 2

Reviewer 1 Report

in revised version, please add first author name instead of saying in Ref [x] this is not good way or citing a paper. 

for example you need to say Author et al. [ ] found that...

Author Response

Dear Reviewer,

We revised the manuscript accordingly. Please see the attachment.
